# Advancing mathematics by guiding human intuition with AI

Alex Davies[1 ✉], Petar Veličković[1], Lars Buesing[1], Sam Blackwell[1], Daniel Zheng[1], Nenad Tomašev[1], Richard Tanburn[1], Peter Battaglia[1], Charles Blundell[1], András Juhász[2], Marc Lackenby[2], Geordie Williamson[3], Demis Hassabis[1] & Pushmeet Kohli[1 ✉]

The practice of mathematics involves discovering patterns and using these to formulate and prove conjectures, resulting in theorems. Since the 1960s, mathematicians have used computers to assist in the discovery of patterns and formulation of conjectures[1], most famously in the Birch and Swinnerton-Dyer conjecture[2], a Millennium Prize Problem[3]. Here we provide examples of new fundamental results in pure mathematics that have been discovered with the assistance of machine learning—demonstrating a method by which machine learning can aid mathematicians in discovering new conjectures and theorems. We propose a process of using machine learning to discover potential patterns and relations between mathematical objects, understanding them with attribution techniques and using these observations to guide intuition and propose conjectures. We outline this machine-learning-guided framework and demonstrate its successful application to current research questions in distinct areas of pure mathematics, in each case showing how it led to meaningful mathematical contributions on important open problems: a new connection between the algebraic and geometric structure of knots, and a candidate algorithm predicted by the combinatorial invariance conjecture for symmetric groups[4]. Our work may serve as a model for collaboration between the fields of mathematics and artificial intelligence (AI) that can achieve surprising results by leveraging the respective strengths of mathematicians and machine learning.

One of the central drivers of mathematical progress is the discovery of patterns and formulation of useful conjectures: statements that are suspected to be true but have not been proven to hold in all cases. Mathematicians have always used data to help in this process—from the early hand-calculated prime tables used by Gauss and others that led to the prime number theorem[5], to modern computer-generated data[1,5] in cases such as the Birch and Swinnerton-Dyer conjecture[2]. The introduction of computers to generate data and test conjectures afforded mathematicians a new understanding of problems that were previously inaccessible[6], but while computational techniques have become consistently useful in other parts of the mathematical process[7,8], artificial intelligence (AI) systems have not yet established a similar place. Prior systems for generating conjectures have either contributed genuinely useful research conjectures[9] via methods that do not easily generalize to other mathematical areas[10], or have demonstrated novel, general methods for finding conjectures[11] that have not yet yielded mathematically valuable results.

AI, in particular the field of machine learning[12–14], offers a collection of techniques that can effectively detect patterns in data and has increasingly demonstrated utility in scientific disciplines[15]. In mathematics, it has been shown that AI can be used as a valuable tool by finding counterexamples to existing conjectures[16], accelerating calculations[17], generating symbolic solutions[18] and detecting the existence of structure in mathematical objects[19]. In this work, we demonstrate that AI can also be used to assist in the discovery of theorems and conjectures at the forefront of mathematical research. This extends work using supervised learning to find patterns[20–24] by focusing on enabling mathematicians to understand the learned functions and derive useful mathematical insight. We propose a framework for augmenting the standard mathematician's toolkit with powerful pattern recognition and interpretation methods from machine learning and demonstrate its value and generality by showing how it led us to two fundamental new discoveries, one in topology and another in representation theory. Our contribution shows how mature machine learning methodologies can be adapted and integrated into existing mathematical workflows to achieve novel results.

## Guiding mathematical intuition with AI

A mathematician's intuition plays an enormously important role in mathematical discovery—"It is only with a combination of both rigorous formalism and good intuition that one can tackle complex mathematical problems"[25]. The following framework, illustrated in Fig. 1, describes a general method by which mathematicians can use tools from machine learning to guide their intuitions concerning complex mathematical objects, verifying their hypotheses about the existence of relationships and helping them understand those relationships. We propose that this is a natural and empirically productive way that these

[1]DeepMind, London, UK. [2]University of Oxford, Oxford, UK. [3]University of Sydney, Sydney, New South Wales, Australia. ✉e-mail: adavies@deepmind.com; pushmeet@deepmind.com

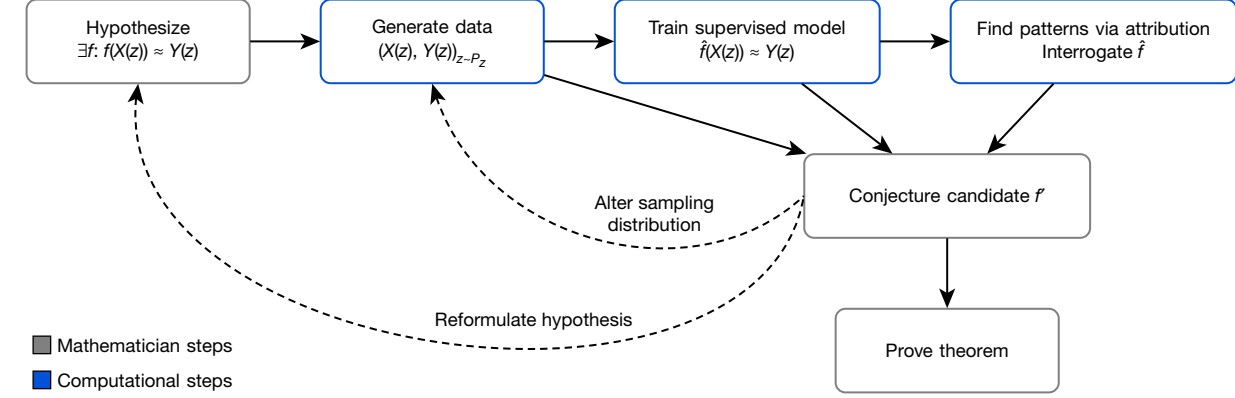

**Fig. 1 | Flowchart of the framework.** The process helps guide a mathematician's intuition about a hypothesized function *f*, by training a machine learning model to estimate that function over a particular distribution of data $P_z$. The insights from the accuracy of the learned function $\hat{f}$ and attribution techniques applied to it can aid in the understanding of the problem and the construction of a closed-form *f'*. The process is iterative and interactive, rather than a single series of steps.

well-understood techniques in statistics and machine learning can be used as part of a mathematician's work.

Concretely, it helps guide a mathematician's intuition about the relationship between two mathematical objects $X(z)$ and $Y(z)$ associated with $z$ by identifying a function $\hat{f}$ such that $\hat{f}(X(z)) \approx Y(z)$ and analysing it to allow the mathematician to understand properties of the relationship. As an illustrative example: let $z$ be convex polyhedra, $X(z) \in \mathbb{Z}^2 \times \mathbb{R}^2$ be the number of vertices and edges of $z$, as well as the volume and surface area, and $Y(z) \in \mathbb{Z}$ be the number of faces of z. Euler's formula states that there is an exact relationship between $X(z)$ and $Y(z)$ in this case: $X(z) \cdot (-1, 1, 0, 0) + 2 = Y(z)$. In this simple example, among many other ways, the relationship could be rediscovered by the traditional methods of data-driven conjecture generation[1]. However, for $X(z)$ and $Y(z)$ in higher-dimensional spaces, or of more complex types, such as graphs, and for more complicated, nonlinear $\hat{f}$, this approach is either less useful or entirely infeasible.

The framework helps guide the intuition of mathematicians in two ways: by verifying the hypothesized existence of structure/patterns in mathematical objects through the use of supervised machine learning; and by helping in the understanding of these patterns through the use of attribution techniques.

In the supervised learning stage, the mathematician proposes a hypothesis that there exists a relationship between $X(z)$ and $Y(z)$. By generating a dataset of $X(z)$ and $Y(z)$ pairs, we can use supervised learning to train a function $\hat{f}$ that predicts $Y(z)$, using only $X(z)$ as input. The key contributions of machine learning in this regression process are the broad set of possible nonlinear functions that can be learned given a sufficient amount of data. If $\hat{f}$ is more accurate than would be expected by chance, it indicates that there may be such a relationship to explore. If so, attribution techniques can help in the understanding of the learned function $\hat{f}$ sufficiently for the mathematician to conjecture a candidate *f'*. Attribution techniques can be used to understand which aspects of $\hat{f}$ are relevant for predictions of $Y(z)$. For example, many attribution techniques aim to quantify which component of $X(z)$ the function $\hat{f}$ is sensitive to. The attribution technique we use in our work, gradient saliency, does this by calculating the derivative of outputs of $\hat{f}$, with respect to the inputs. This allows a mathematician to identify and prioritize aspects of the problem that are most likely to be relevant for the relationship. This iterative process might need to be repeated several times before a viable conjecture is settled on. In this process, the mathematician can guide the choice of conjectures to those that not just fit the data but also seem interesting, plausibly true and, ideally, suggestive of a proof strategy.

Conceptually, this framework provides a 'test bed for intuition'— quickly verifying whether an intuition about the relationship between two quantities may be worth pursuing and, if so, guidance as to how they may be related. We have used the above framework to help mathematicians to obtain impactful mathematical results in two cases—discovering and proving one of the first relationships between algebraic and geometric invariants in knot theory and conjecturing a resolution to the combinatorial invariance conjecture for symmetric groups[4], a well-known conjecture in representation theory. In each area, we demonstrate how the framework has successfully helped guide the mathematician to achieve the result. In each of these cases, the necessary models can be trained within several hours on a machine with a single graphics processing unit.

## Topology

Low-dimensional topology is an active and influential area of mathematics. Knots, which are simple closed curves in $\mathbb{R}^3$, are one of the key objects that are studied, and some of the subject's main goals are to classify them, to understand their properties and to establish connections with other fields. One of the principal ways that this is carried out is through invariants, which are algebraic, geometric or numerical quantities that are the same for any two equivalent knots. These invariants are derived in many different ways, but we focus on two of the main categories: hyperbolic invariants and algebraic invariants. These two types of invariants are derived from quite different mathematical disciplines, and so it is of considerable interest to establish connections between them. Some examples of these invariants for small knots are shown in Fig. 2. A notable example of a conjectured connection is the volume conjecture[26], which proposes that the hyperbolic volume of a knot (a geometric invariant) should be encoded within the asymptotic behaviour of its coloured Jones polynomials (which are algebraic invariants).

Our hypothesis was that there exists an undiscovered relationship between the hyperbolic and algebraic invariants of a knot. A supervised learning model was able to detect the existence of a pattern between a large set of geometric invariants and the signature $\sigma(K)$, which is known to encode important information about a knot $K$, but was not previously known to be related to the hyperbolic geometry. The most relevant features identified by the attribution technique, shown in Fig. 3a, were three invariants of the cusp geometry, with the relationship visualized partly in Fig. 3b. Training a second model with $X(z)$ consisting of only these measurements achieved a very similar accuracy, suggesting that they are a sufficient set of features to capture almost all of the effect of the geometry on the signature. These three invariants were the real and imaginary parts of the meridional translation $\mu$ and the longitudinal translation $\lambda$. There is a nonlinear, multivariate relationship between these quantities and the signature. Having been guided to focus on these invariants, we discovered that this relationship is best understood by means of a new quantity, which is linearly related to the signature.

| z: **Knot** | X(z): **Geometric invariants** | | | Y(z): **Algebraic invariants** | | |
| --- | --- | --- | --- | --- | --- | --- |
| | Volume | Chern–Simons | Meridional translation | ... | Signature | Jones polynomial | ... |
| | 2.0299 | 0 | $i$ | ... | 0 | $t^{-2} - t^{-1} + 1 - t + t^2$ | ... |
| | 2.8281 | −0.1532 | $0.7381 + 0.8831i$ | ... | −2 | $t - t^2 + 2t^3 - t^4 + t^5 - t^6$ | ... |
| | 3.1640 | 0.1560 | $-0.7237 + 1.0160i$ | ... | 0 | $t^{-2} - t^{-1} + 2 - 2t + t^2 - t^3 + t^4$ | ... |

**Fig. 2 | Examples of invariants for three hyperbolic knots.** We hypothesized that there was a previously undiscovered relationship between the geometric and algebraic invariants.

We introduce the 'natural slope', defined to be slope($K$) = Re($\lambda/\mu$), where Re denotes the real part. It has the following geometric interpretation. One can realize the meridian curve as a geodesic $\gamma$ on the Euclidean torus. If one fires off a geodesic $\gamma^\perp$ from this orthogonally, it will eventually return and hit $\gamma$ at some point. In doing so, it will have travelled along a longitude minus some multiple of the meridian. This multiple is the natural slope. It need not be an integer, because the endpoint of $\gamma^\perp$ might not be the same as its starting point. Our initial conjecture relating natural slope and signature was as follows.

Conjecture: There exist constants $c_1$ and $c_2$ such that, for every hyperbolic knot $K$,

$$|2\sigma(K) - \text{slope}(K)| < c_1\text{vol}(K) + c_2 \quad (1)$$

While this conjecture was supported by an analysis of several large datasets sampled from different distributions, we were able to construct counterexamples using braids of a specific form. Subsequently, we were able to establish a relationship between slope($K$), signature $\sigma(K)$, volume vol($K$) and one of the next most salient geometric invariants, the injectivity radius inj($K$) (ref. [27]).

Theorem: There exists a constant $c$ such that, for any hyperbolic knot $K$,

$$|2\sigma(K) - \text{slope}(K)| \le c\,\text{vol}(K)\text{inj}(K)^{-3} \quad (2)$$

It turns out that the injectivity radius tends not to get very small, even for knots of large volume. Hence, the term inj($K$)$^{-3}$ tends not to get very large in practice. However, it would clearly be desirable to have a theorem that avoided the dependence on inj($K$)$^{-3}$, and we give such a result that instead relies on short geodesics, another of the most salient features, in the Supplementary Information. Further details and a full proof of the above theorem are available in ref. [27]. Across the datasets we generated, we can place a lower bound of $c \ge 0.23392$, and it would be reasonable to conjecture that $c$ is at most 0.3, which gives a tight relationship in the regions in which we have calculated.

The above theorem is one of the first results that connect the algebraic and geometric invariants of knots and has various interesting applications. It directly implies that the signature controls the non-hyperbolic Dehn surgeries on the knot and that the natural slope controls the genus of surfaces in $\mathbb{R}_+^4$ whose boundary is the knot. We expect that this newly discovered relationship between natural slope and signature will have many other applications in low-dimensional topology. It is surprising that a simple yet profound connection such as this has been overlooked in an area that has been extensively studied.

## Representation theory

Representation theory is the theory of linear symmetry. The building blocks of all representations are the irreducible ones, and understanding them is one of the most important goals of representation theory. Irreducible representations generalize the fundamental frequencies of Fourier analysis[28]. In several important examples, the structure of irreducible representations is governed by Kazhdan–Lusztig (KL) polynomials, which have deep connections to combinatorics, algebraic geometry and singularity theory. KL polynomials are polynomials attached to pairs of elements in symmetric groups (or more generally, pairs of elements in

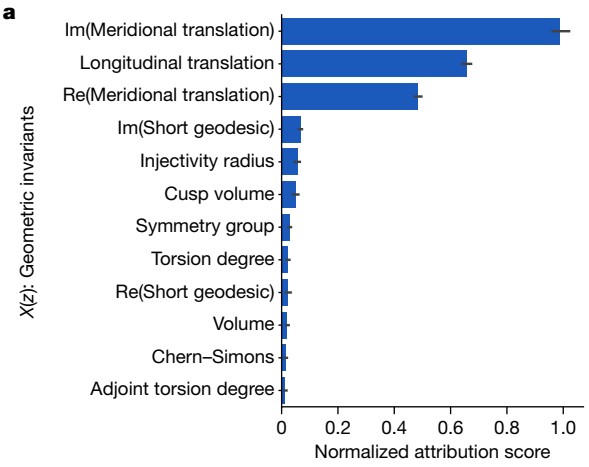

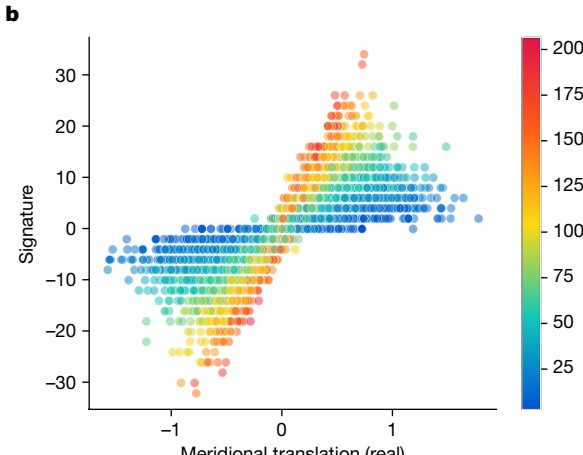

**Fig. 3 | Knot theory attribution. a**, Attribution values for each of the input X(z). The features with high values are those that the learned function is most sensitive to and are probably relevant for further exploration. The 95%

confidence interval error bars are across 10 retrainings of the model.
**b**, Example visualization of relevant features—the real part of the meridional translation against signature, coloured by the longitudinal translation.

| $z$: **Pair of permutations** | $X(z)$: **Unlabelled Bruhat interval** | $Y(z)$: **KL polynomial** |
|---|---|---|
| (03214), (34201) | | $1 + q^2$ |
| (021435), (240513) | | $1 + 2q + q^2$ |

**Fig. 4 | Two example dataset elements, one from $S_5$ and one from $S_6$.** The combinatorial invariance conjecture states that the KL polynomial of a pair of permutations should be computable from their unlabelled Bruhat interval, but no such function was previously known.

Coxeter groups). The combinatorial invariance conjecture is a fascinating open conjecture concerning KL polynomials that has stood for 40 years, with only partial progress[29]. It states that the KL polynomial of two elements in a symmetric group $S_N$ can be calculated from their unlabelled Bruhat interval[30], a directed graph. One barrier to progress in understanding the relationship between these objects is that the Bruhat intervals for non-trivial KL polynomials (those that are not equal to 1) are very large graphs that are difficult to develop intuition about. Some examples of small Bruhat intervals and their KL polynomials are shown in Fig. 4.

We took the conjecture as our initial hypothesis, and found that a supervised learning model was able to predict the KL polynomial from the Bruhat interval with reasonably high accuracy. By experimenting on the way in which we input the Bruhat interval to the network, it became apparent that some choices of graphs and features were particularly conducive to accurate predictions. In particular, we found that a subgraph inspired by prior work[31] may be sufficient to calculate the KL polynomial, and this was supported by a much more accurate estimated function.

Further structural evidence was found by calculating salient subgraphs that attribution techniques determined were most relevant and analysing the edge distribution in these graphs compared to the original graphs. In Fig. 5a, we aggregate the relative frequency of the edges in the salient subgraphs by the reflection that they represent. It shows that extremal reflections (those of the form $(0, i)$ or $(i, N-1)$ for $S_N$) appear more commonly in salient subgraphs than one would expect, at the expense of simple reflections (those of the form $(i, i+1)$), which is confirmed over many retrainings of the model in Fig. 5b. This is notable because the edge labels are not given to the network and are not recoverable from the unlabelled Bruhat interval. From the definition of KL polynomials, it is intuitive that the distinction between simple and non-simple reflections is relevant for calculating it; however, it was not initially obvious why extremal reflections would be overrepresented in salient subgraphs. Considering this observation led us to the discovery that there is a natural decomposition of an interval into two parts—a hypercube induced by one set of extremal edges and a graph isomorphic to an interval in $S_{N-1}$.

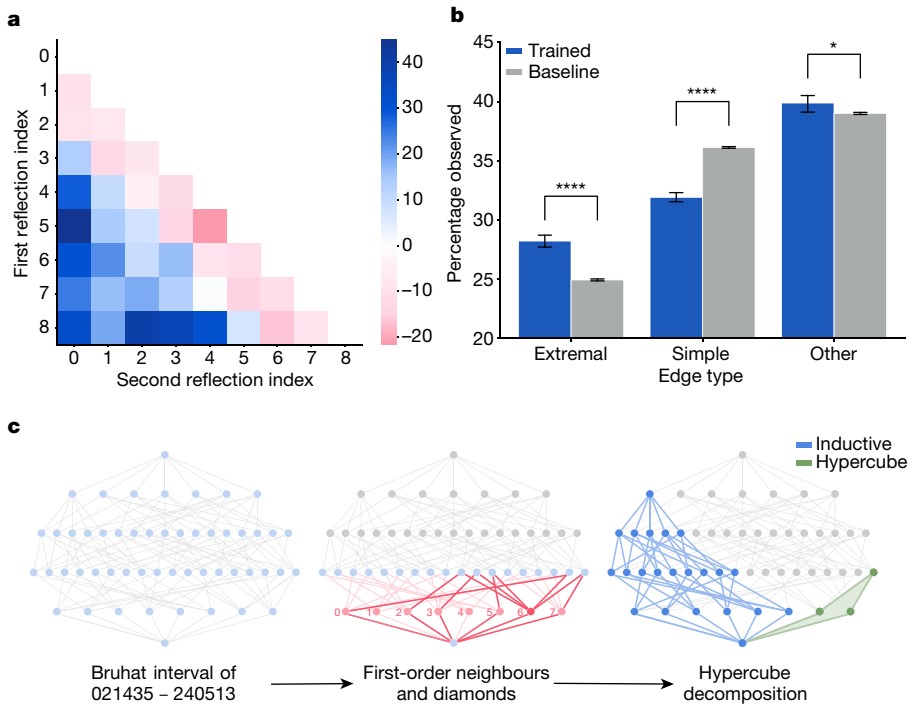

**Fig. 5 | Representation theory attribution. a**, An example heatmap of the percentage increase in reflections present in the salient subgraphs compared with the average across intervals in the dataset when predicting $q^4$. **b**, The percentage of observed edges of each type in the salient subgraph for 10 retrainings of the model compared to 10 bootstrapped samples of the same size from the dataset. The error bars are 95% confidence intervals, and the significance level shown was determined using a two-sided two-sample $t$-test. *$p < 0.05$; ****$p < 0.0001$. **c**, Illustration for the interval $021435–240513 \in S_6$ of the interesting substructures that were discovered through the iterative process of hypothesis, supervised learning and attribution. The subgraph inspired by previous work[31] is highlighted in red, the hypercube in green and the decomposition component isomporphic to an interval in $S_{N-1}$ in blue.

# Article

The importance of these two structures, illustrated in Fig. 5c, led to a proof that the KL polynomial can be computed directly from the hypercube and $S_{N-1}$ components through a beautiful formula that is summarized in the Supplementary Information. A further detailed treatment of the mathematical results is given in ref. [32].

**Theorem:** Every Bruhat interval admits a canonical hypercube decomposition along its extremal reflections, from which the KL polynomial is directly computable.

Remarkably, further tests suggested that all hypercube decompositions correctly determine the KL polynomial. This has been computationally verified for all of the $\sim 3 \times 10^6$ intervals in the symmetric groups up to $S_7$ and more than $1.3 \times 10^5$ non-isomorphic intervals sampled from the symmetric groups $S_8$ and $S_9$.

**Conjecture:** The KL polynomial of an unlabelled Bruhat interval can be calculated using the previous formula with any hypercube decomposition.

This conjectured solution, if proven true, would settle the combinatorial invariance conjecture for symmetric groups. This is a promising direction as not only is the conjecture empirically verified up to quite large examples, but it also has a particularly nice form that suggests potential avenues for attacking the conjecture. This case demonstrates how non-trivial insights about the behaviour of large mathematical objects can be obtained from trained models, such that new structure can be discovered.

## Conclusion

In this work we have demonstrated a framework for mathematicians to use machine learning that has led to mathematical insight across two distinct disciplines: one of the first connections between the algebraic and geometric structure of knots and a proposed resolution to a long-standing open conjecture in representation theory. Rather than use machine learning to directly generate conjectures, we focus on helping guide the highly tuned intuition of expert mathematicians, yielding results that are both interesting and deep. It is clear that intuition plays an important role in elite performance in many human pursuits. For example, it is critical for top Go players and the success of AlphaGo (ref. [33]) came in part from its ability to use machine learning to learn elements of play that humans perform intuitively. It is similarly seen as critical for top mathematicians—Ramanujan was dubbed the Prince of Intuition[34] and it has inspired reflections by famous mathematicians on its place in their field[35,36]. As mathematics is a very different, more cooperative endeavour than Go, the role of AI in assisting intuition is far more natural. Here we show that there is indeed fruitful space to assist mathematicians in this aspect of their work.

Our case studies demonstrate how a foundational connection in a well-studied and mathematically interesting area can go unnoticed, and how the framework allows mathematicians to better understand the behaviour of objects that are too large for them to otherwise observe patterns in. There are limitations to where this framework will be useful—it requires the ability to generate large datasets of the representations of objects and for the patterns to be detectable in examples that are calculable. Further, in some domains the functions of interest may be difficult to learn in this paradigm. However, we believe there are many areas that could benefit from our methodology. More broadly, it is our hope that this framework is an effective mechanism to allow for the introduction of machine learning into mathematicians' work, and encourage further collaboration between the two fields.

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

## Methods

### Framework

**Supervised learning.** In the supervised learning stage, the mathematician proposes a hypothesis that there exists a relationship between $X(z)$ and $Y(z)$. In this work we assume that there is no known function mapping from $X(z)$ to $Y(z)$, which in turn implies that $X$ is not invertible (otherwise there would exist a known function $Y \circ X^{-1}$). While there may still be value to this process when the function is known, we leave this for future work. To test the hypothesis that $X$ and $Y$ are related, we generate a dataset of $X(z)$, $Y(z)$ pairs, where $z$ is sampled from a distribution $P_Z$. The results of the subsequent stages will hold true only for the distribution $P_Z$, and not the whole space $Z$. Initially, sensible choices for $P_Z$ would be, for example, uniformly over the first $N$ items for $Z$ with a notion of ordering, or uniformly at random where possible. In subsequent iterations, $P_Z$ may be chosen to understand the behaviour on different parts of the space $Z$ (for example, regions of $Z$ that may be more likely to provide counterexamples to a particular hypothesis). To first test whether a relation between $X(z)$ and $Y(z)$ can be found, we use supervised learning to train a function $\hat{f}$ that approximately maps $X(z)$ to $Y(z)$. In this work we use neural networks as the supervised learning method, in part because they can be easily adapted to many different types of $X$ and $Y$ and knowledge of any inherent geometry (in terms of invariances and symmetries) of the input domain $X$ can be incorporated into the architecture of the network[37]. We consider a relationship between $X(z)$ and $Y(z)$ to be found if the accuracy of the learned function $\hat{f}$ is statistically above chance on further samples from $P_Z$ on which the model was not trained. The converse is not true; namely, if the model cannot predict the relationship better than chance, it may mean that a pattern exists, but is sufficiently complicated that it cannot be captured by the given model and training procedure. If it does indeed exist, this can give a mathematician confidence to pursue a particular line of enquiry in a problem that may otherwise be only speculative.

**Attribution techniques.** If a relationship is found, the attribution stage is to probe the learned function $\hat{f}$ with attribution techniques to further understand the nature of the relationship. These techniques attempt to explain what features or structures are relevant to the predictions made by $\hat{f}$, which can be used to understand what parts of the problem are relevant to explore further. There are many attribution techniques in the body of literature on machine learning and statistics, including stepwise forward feature selection[38], feature occlusion and attention weights[39]. In this work we use gradient-based techniques[40], broadly similar to sensitivity analysis in classical statistics and sometimes referred to as saliency maps. These techniques attribute importance to the elements of $X(z)$, by calculating how much $\hat{f}$ changes in predictions of $Y(z)$ given small changes in $X(z)$. We believe these are a particularly useful class of attribution techniques as they are conceptually simple, flexible and easy to calculate with machine learning libraries that support automatic differentiation[41–43]. Information extracted via attribution techniques can then be useful to guide the next steps of mathematical reasoning, such as conjecturing closed-form candidates $f'$, altering the sampling distribution $P_Z$ or generating new hypotheses about the object of interest $z$, as shown in Fig. 1. This can then lead to an improved or corrected version of the conjectured relationship between these quantities.

### Topology

**Problem framing.** Not all knots admit a hyperbolic geometry; however, most do, and all knots can be constructed from hyperbolic and torus knots using satellite operations[44]. In this work we focus only on hyperbolic knots. We characterize the hyperbolic structure of the knot complement by a number of easily computable invariants. These invariants do not fully define the hyperbolic structure, but they are representative of the most commonly interesting properties of the geometry. Our

initial general hypothesis was that the hyperbolic invariants would be predictive of algebraic invariants. The specific hypothesis we investigated was that the geometry is predictive of the signature. The signature is an ideal candidate as it is a well-understood and common invariant, it is easy to calculate for large knots and it is an integer, which makes the prediction task particularly straightforward (compared to, for example, a polynomial).

**Data generation.** We generated a number of datasets from different distributions $P_Z$ on the set of knots using the SnapPy software package[45], as follows.

1. All knots up to 16 crossings ($\sim 1.7 \times 10^6$ knots), taken from the Regina census[46].
2. Random knot diagrams of 80 crossings generated by SnapPy's random_link function ($\sim 10^6$ knots). As random knot generation can potentially lead to duplicates, we calculate a large number of invariants for each knot diagram and remove any samples that have identical invariants to a previous sample, as they are likely to represent that same knot with very high probability.
3. Knots obtained as the closures of certain braids. Unlike the previous two datasets, the knots that were produced here are not, in any sense, generic. Instead, they were specifically constructed to disprove Conjecture 1. The braids that we used were 4-braids ($n = 11{,}756$), 5-braids ($n = 13{,}217$) and 6-braids ($n = 10{,}897$). In terms of the standard generators $\sigma_i$ for these braid groups, the braids were chosen to be $(\sigma_{i_1}^{n_1} \sigma_{i_2}^{n_2} \dots \sigma_{i_k}^{n_k})^N$. The integers $i_j$ were chosen uniformly at random for the appropriate braid group. The powers $n_j$ were chosen uniformly at random in the ranges $[-10, -3]$ and $[3, 10]$. The final power $N$ was chosen uniformly between 1 and 10. The quantity $\sum |n_i|$ was restricted to be at most 15 for 5-braids and 6-braids and 12 for 4-braids, and the total number of crossings $N \sum |n_i|$ was restricted to lie in the range between 10 and 60. The rationale for these restrictions was to ensure a rich set of examples that were small enough to avoid an excessive number of failures in the invariant computations.

For the above datasets, we computed a number of algebraic and geometric knot invariants. Different datasets involved computing different subsets of these, depending on their role in forming and examining the main conjecture. Each of the datasets contains a subset of the following list of invariants: signature, slope, volume, meridional translation, longitudinal translation, injectivity radius, positivity, Chern–Simons invariant, symmetry group, hyperbolic torsion, hyperbolic adjoint torsion, invariant trace field, normal boundary slopes and length spectrum including the linking numbers of the short geodesics.

The computation of the canonical triangulation of randomly generated knots fails in SnapPy in our data generation process in between 0.6% and 1.7% of the cases, across datasets. The computation of the injectivity radius fails between 2.8% of the time on smaller knots up to 7.8% of the time on datasets of knots with a higher number of crossings. On knots up to 16 crossings from the Regina dataset, the injectivity radius computation failed in 5.2% of the cases. Occasional failures can occur in most of the invariant computations, in which case the computations continue for the knot in question for the remaining invariants in the requested set. Additionally, as the computational complexity of some invariants is high, operations can time out if they take more than 5 min for an invariant. This is a flexible bound and ultimately a trade-off that we have used only for the invariants that were not critical for our analysis, to avoid biasing the results.

**Data encoding.** The following encoding scheme was used for converting the different types of features into real valued inputs for the network: reals directly encoded; complex numbers as two reals corresponding to the real and imaginary parts; categoricals as one-hot vectors.

All features are normalized by subtracting the mean and dividing by the variance. For simplicity, in Fig. 3a, the salience values of categoricals

are aggregated by taking the maximum value of the saliencies of their encoded features.

**Model and training procedure.** The model architecture used for the experiments was a fully connected, feed-forward neural network, with hidden unit sizes [300, 300, 300] and sigmoid activations. The task was framed as a multi-class classification problem, with the distinct values of the signature as classes, cross-entropy loss as an optimizable loss function and test classification accuracy as a metric of performance. It is trained for a fixed number of steps using a standard optimizer (Adam). All settings were chosen as a priori reasonable values and did not need to be optimized.

**Process.** First, to assess whether there may be a relationship between the geometry and algebra of a knot, we trained a feed-forward neural network to predict the signature from measurements of the geometry on a dataset of randomly sampled knots. The model was able to achieve an accuracy of 78% on a held-out test set, with no errors larger than ±2. This is substantially higher than chance (a baseline accuracy of 25%), which gave us strong confidence that a relationship may exist.

To understand how this prediction is being made by the network, we used gradient-based attribution to determine which measurements of the geometry are most relevant to the signature. We do this using a simple sensitivity measure $r_i$ that averages the gradient of the loss $L$ with respect to a given input feature $x_i$ over all of the examples $x$ in a dataset $\mathcal{X}$:

$$r_i = \frac{1}{|\mathcal{X}|} \sum_{x \in \mathcal{X}} \left| \frac{\partial L}{\partial x_i} \right| \tag{3}$$

This quantity for each input feature is shown in Fig. 3a, where we can determine that the relevant measurements of the geometry appear to be what is known as the cusp shape: the meridional translation, which we will denote $\mu$, and the longitudinal translation, which we will denote $\lambda$. This was confirmed by training a new model to predict the signature from only these three measurements, which was able to achieve the same level of performance as the original model.

To confirm that the slope is a sufficient aspect of the geometry to focus on, we trained a model to predict the signature from the slope alone. Visual inspection of the slope and signature in Extended Data Fig. 1a, b shows a clear linear trend, and training a linear model on this data results in a test accuracy of 78%, which is equivalent to the predictive power of the original model. This implies that the slope linearly captures all of the information about the signature that the original model had extracted from the geometry.

**Evaluation.** The confidence intervals on the feature saliencies were calculated by retraining the model 10 times with a different train/test split and a different random seed initializing both the network weights and training procedure.

### Representation theory
**Data generation.** For our main dataset we consider the symmetric groups up to $S_9$. The first symmetric group that contains a non-trivial Bruhat interval whose KL polynomial is not simply 1 is $S_5$, and the largest interval in $S_9$ contains $9! \approx 3.6 \times 10^5$ nodes, which starts to pose computational issues when used as inputs to networks. The number of intervals in a symmetric group $S_N$ is $O(N!^2)$, which results in many billions of intervals in $S_9$. The distribution of coefficients of the KL polynomials uniformly across intervals is very imbalanced, as higher coefficients are especially rare and associated with unknown complex structure. To adjust for this and simplify the learning problem, we take advantage of equivalence classes of Bruhat intervals that eliminate many redundant small polynomials[47]. This has the added benefit of reducing the number of intervals per symmetric group

(for example, to ~2.9 million intervals in $S_9$). We further reduce the dataset by including a single interval for each distinct KL polynomial for all graphs with the same number of nodes, resulting in 24,322 non-isomorphic graphs for $S_9$. We split the intervals randomly into train/test partitions at 80%/20%.

**Data encoding.** The Bruhat interval of a pair of permutations is a partially ordered set of the elements of the group, and it can be represented as a directed acyclic graph where each node is labelled by a permutation, and each edge is labelled by a reflection. We add two features at each node representing the in-degree and out-degree of that node.

**Model and training procedure.** For modelling the Bruhat intervals, we used a particular GraphNet architecture called a message-passing neural network (MPNN)[48]. The design of the model architecture (in terms of activation functions and directionality) was motivated by the algorithms for computing KL polynomials from labelled Bruhat intervals. While labelled Bruhat intervals contain privileged information, these algorithms hinted at the kind of computation that may be useful for computing KL polynomial coefficients. Accordingly, we designed our MPNN to algorithmically align to this computation[49]. The model is bi-directional, with a hidden layer width of 128, four propagation steps and skip connections. We treat the prediction of each coefficient of the KL polynomial as a separate classification problem.

**Process.** First, to gain confidence that the conjecture is correct, we trained a model to predict coefficients of the KL polynomial from the unlabelled Bruhat interval. We were able to do so across the different coefficients with reasonable accuracy (Extended Data Table 1) giving some evidence that a general function may exist, as a four-step MPNN is a relatively simple function class. We trained a GraphNet model on the basis of a newly hypothesized representation and could achieve significantly better performance, lending evidence that it is a sufficient and helpful representation to understand the KL polynomial.

To understand how the predictions were being made by the learned function $\hat{f}$, we used gradient-based attribution to define a salient subgraph $S_G$ for each example interval $G$, induced by a subset of nodes in that interval, where $L$ is the loss and $x_v$ is the feature for vertex $v$:

$$S_G = \left\{ v \in G \left| \left\| \frac{\partial L}{\partial x_v} \right\| > C_k \right. \right\} \tag{4}$$

We then aggregated the edges by their edge type (each is a reflection) and compared the frequency of their occurrence to the overall dataset. The effect on extremal edges was present in the salient subgraphs for predictions of the higher-order terms ($q^3$, $q^4$), which are the more complicated and less well-understood terms.

**Evaluation.** The threshold $C_k$ for salient nodes was chosen a priori as the 99th percentile of attribution values across the dataset, although the results are present for different values of $C_k$ in the range [95, 99.5]. In Fig. 5a, we visualize a measure of edge attribution for a particular snapshot of a trained model for expository purposes. This view will change across time and random seeds, but we can confirm that the pattern remains by looking at aggregate statistics over many runs of training the model, as in Fig. 5b. In this diagram, the two-sample two-sided $t$-test statistics are as follows—simple edges: $t = 25.7$, $P = 4.0 \times 10^{-10}$; extremal edges: $t = -13.8$, $P = 1.1 \times 10^{-7}$; other edges: $t = -3.2$, $P = 0.01$. These significance results are robust to different settings of the hyper-parameters of the model.

## Code availability
Interactive notebooks to regenerate the results for both knot theory and representation theory have been made available for download at https://github.com/deepmind.

## Data availability

The generated datasets used in the experiments have been made available for download at https://github.com/deepmind.

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

**Acknowledgements** We thank J. Ellenberg, S. Mohamed, O. Vinyals, A. Gaunt, A. Fawzi and D. Saxton for advice and comments on early drafts; J. Vonk for contemporary supporting work; X. Glorot and M. Overlan for insight and assistance; and A. Pierce, N. Lambert, G. Holland, R. Ahamed and C. Meyer for assistance coordinating the research. This research was funded by DeepMind.

**Author contributions** A.D., D.H. and P.K. conceived of the project. A.D., A.J. and M.L. discovered the knot theory results, with D.Z. and N.T. running additional experiments. A.D., P.V. and G.W. discovered the representation theory results, with P.V. designing the model, L.B. running additional experiments, and C.B. providing advice and ideas. S.B. and R.T. provided additional support, experiments and infrastructure. A.D., D.H. and P.K. directed and managed the project. A.D. and P.V. wrote the paper with help and feedback from P.B., C.B., M.L., A.J., G.W., P.K. and D.H.

**Competing interests** The authors declare no competing interests.

**Additional information**
**Correspondence and requests for materials** should be addressed to Alex Davies or Pushmeet Kohli.

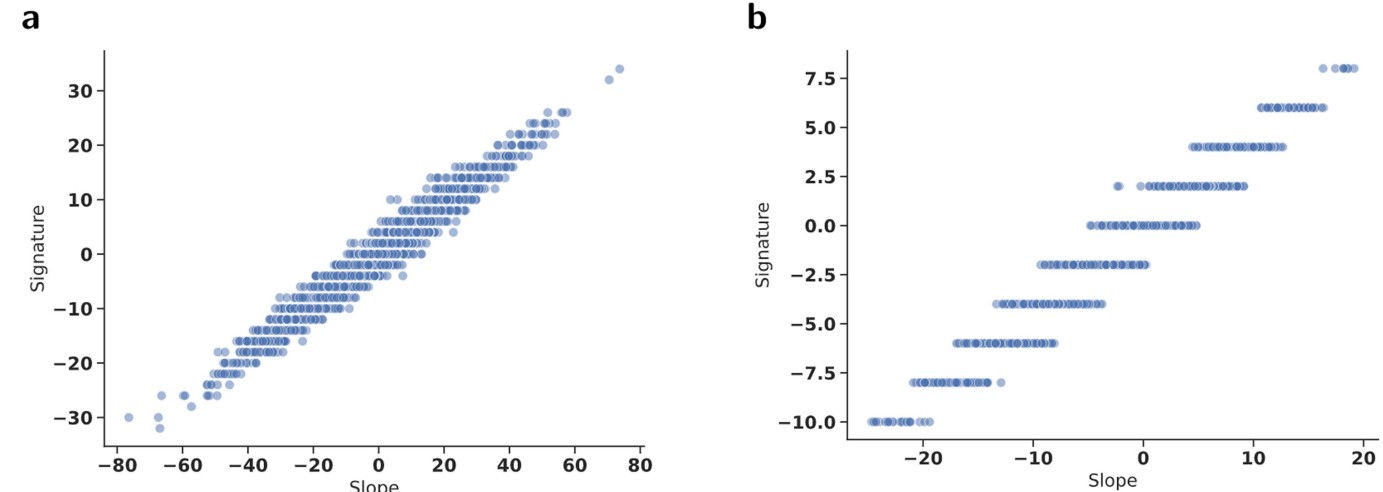

**Extended Data Fig. 1 | Empirical relationship between slope and signature. a** Signature vs slope for random dataset. **b** Signature vs slope for Regina dataset.

**Extended Data Table 1 | Model accuracies at predicting KL coefficients from Bruhat intervals in S$_9$**

|  | $q$ | $q^2$ | $q^3$ | $q^4$ |
|---|---|---|---|---|
| Baseline accuracy | 21% | 12% | 29% | 88% |
| Full interval test accuracy | 98% | 63% | 72% | 98% |
| Dihedral annotated test accuracy | 99.9% | 96.5% | 95.6% | 99.4% |