## [Peer Review File · Nature]

Manuscript Title: Advancing mathematics by guiding human intuition with AI

Reviewer Comments & Author Rebuttals

Reviewer Reports on the Initial Version:

Referee #1 (Remarks to the Author):

As requested by the editor, my primary role as a referee is to review the significance of the new result in knot theory. For this, I looked at both the main article and the two supplementary files.

The new result, which relates the geometrically defined "natural slope" to the purely topological "signature" is indeed quite striking and unexpected. Independent of its unusual origin, I judge the separate mathematical paper "The signature and cusp geometry of hyperbolic knots" likely to be accepted by the top speciality journal in the field (Geometry and Topology) and be a shoo in at the second best one (J. Topology). For scale, only the top general math journals Annals, JAMS, and Inventiones, which are analogous in math to Nature and Science, have higher standards than Geometry and Topology. So to me the machine learning method of the main paper is an exciting new tool given that it can lead to new results of this quality!

I have two more detailed comments:

1. The initial connection observed between slope and signature is a linear relationship which is extremely marked, as shown in Figure 2(b) of the main submission and especially Figure 5 of "Supplementary Material: The signature and cusp shape of hyperbolic knots". It would strengthen the case for using machine learning if you explained why this wouldn't have been noticed by a quick correlation check of the variables or a naive linear regression.
2. It might be worth mentioning some prior instances where people have experimentally explored connections between geometric and topological invariants of knots, specifically:

Khovanov, Patterns in Knot Cohomology, I.
<http://projecteuclid.org/euclid.em/1087329238>

Jejja et al, Deep learning the hyperbolic volume of a knot
<https://doi.org/10.1016/j.physletb.2019.135033>

Unlike the current paper, I believe these only led to some interesting (and still unexplained) patterns and not a new theorem as in the paper under review, which highlights the strength of the new method.

Referee #2 (Remarks to the Author):

A Summary of the key results) Summary of paper: It tries to enhance and formalize the idea of discovering new mathematical relationships via computer experimentation. For a mathematical object X (which is a vector of all known attributes or quantitative facts about the object) we wish to understand a quantity $Y(X)$ and understand how it relates to various parts of X . The setup requires some way to generate objects X and compute $Y(X)$. The key ingredients are: (a) using examples of $(X, Y(X))$ learn via supervised machine learning a way to predict $Y(X)$ from X (b) use saliency/attribution methods to learn portions of X that determine $Y(X)$ and (c) derive --again via machine learning-- a relationship between these. (Aside: maybe the methodology should include (c) after line 47?)

At first sight one might fear that if one is able to compute $Y(X)$ already via a separate algorithm then perhaps one already understands $Y(\cdot)$ pretty well already? If so, how interesting would be the new insights? However, the authors present two examples of new relationships found in two areas of mathematics (knot theory and representation theory) via this method which --I presume-- were surprising to the mathematicians.

(B Originality and significance) Given the increasing interest in use of machine learning (and more general computer science ideas) to advancing mathematics, it is timely and significant.

In terms of originality, this project clearly required close collaboration between experts from machine learning and mathematics and I am not aware of any other competing results or efforts.

(C Data & methodology) The paper is well-written and the techniques are sound and of general interest. I verified the details of machine learning and they seem solid.

The paper opens the door for more work of this type. The machine learning component is fairly simple and probably can be carried out by mathematicians who know any sort of computer programming.

(D Appropriate use of statistics and treatment of uncertainties) N/A. (There are no statistical theory applicable to this merger of machine learning and human intuition.)

(E Conclusions: robustness, validity, reliability) See above

(F Suggested improvements: experiments, data for possible revision) Suggestions:

1) The main paper could give a bit of insight to the average reader about attribution methods, since the math is fairly simple. (Namely, Gradient captures how $Y(\cdot)$ changes as we change coordinates of X). Maybe modify the opening example of Euler's formula, but assume X contains "irrelevant" information about the polyhedron. Then the method has to first infer that only number of vertices and edges is relevant information.

2) Most readers will not be experts in knot theory and representation theory, and it would be helpful to indicate how interesting the discovered facts are for experts in these areas. Are these something that an average grad student could have discovered? How about an average math professor?

(3) Include other interesting settings where $Y(X)$ is computable from X , but we don't fully understand function $Y(\cdot)$. Since the author list includes some top mathematicians I assume there have been internal discussions of this issue.

(G References: appropriate credit to previous work?) Reference list is good. It might be helpful to briefly mention various ways in which computer science methods are being applied to mathematics and to mention how this work does not fit in those traditions (eg formal verification).

(H Clarity and context: lucidity of abstract/summary, appropriateness of abstract, introduction and conclusions) See above.

Overall I am positive about the paper.

Referee #3 (Remarks to the Author):

Review of

Advancing mathematics by guiding human intuition with AI

by A. Davies and al.

As I am not an expert in knot theory, nor in AI, I will comment mainly on the representation theory part of the work. I find no flaws in the manuscript which should prohibit its publication. The mathematical conclusions of this work are original. The representation theory results presented are of immediate interest to many people in representation theory, but also in algebraic geometry and topology, given that Kazhdan-Lusztig polynomials play such a central role in representation theory, but also have applications to the algebraic geometry and topology of Schubert varieties (see, e.g., [A. Bjorner, F. Brenti, *Combinatorics of Coxeter Groups*, Graduate Texts in Mathematics, 231, Springer-Verlag, New York, 2005], [S. Billey, V. Lakshmibai, *Singular loci of Schubert varieties*, Progress in Mathematics, 182, Birkhäuser Boston, Inc., Boston, MA, 2000], [J. E. Humphreys, *Representations of semisimple Lie algebras in the BGG category O*, Graduate Studies in Mathematics, vol. 94, Amer. Math. Soc., Providence, RI, xvi+289 pp., 2008], and the references cited there) and that the conjecture studied (the combinatorial invariance conjecture) is now probably the most outstanding open problem about these polynomials, and has been open for about 40 years ([31]). The methodology presented of using AI in mathematical research is of immediate interest to any mathematician who works with mathematical objects that can be conveniently stored and analyzed by a computer. This includes mathematicians working, for example, in combinatorics, algebra, algebraic geometry, and many parts of topology and mathematical physics.

I cannot imagine such a mathematician not using these methods, where available, given that not using them means progressing much more slowly and being unable to analyze very complicated objects and very large datasets. I cannot comment on the validity of the AI method. However, the mathematical results obtained with it are quite impressive, which makes me think that the method is definitely very valid. The authors have done an exceedingly good job of presenting the material to a general scientific audience. The whole paper, including the abstract, is clear and understandable, with a few exceptions noted below. The reporting of data and methodology is sufficiently detailed for a computer scientist to reproduce the results, but I strongly suggest adding quite a few more details (at least a few lines) for a mathematician to be able to do so. Error bars in graphs and tables are accurately described and easy to understand. As I am not an expert in statistics, I cannot comment on the appropriateness of the statistical tests performed. I think that the conclusion of the paper, namely that AI can be an extremely useful tool in mathematical research, is valid and has been very convincingly demonstrated in this work.

This paper marks the beginning of a new phase in the use of computers in mathematical research. While up to now computers have been used to test existing conjectures or, lately, to find new conjectures, this is the first time that computers have been able to suggest avenues of proof for existing conjectures. This has been achieved by allowing humans and deep learning algorithms to interact in a new way, which I consider to be one of the two outstanding features of this work. The other outstanding feature is the quality, novelty, depth, and interest of the conjectures and theorems produced with this method. In particular, the feeling that extremal reflections seem to play a crucial role in the conjectured combinatorial invariance of KL polynomials is one that I would have expected only from a handful of great world experts in representation theory, and is such a "human" suggestion that it gives me the goose bumps. For all the above reasons, I warmly recommend the publication of this paper in *Nature*. The authors conclude their manuscript with the sentence: "More broadly, it is our hope that this framework is an effective mechanism to allow for the introduction of machine learning into mathematicians' work, and encourage further collaboration between the two fields." I can't but join them in their hope.

Detailed comments:

- l. 10 "in ways that do not easily generalize". I feel that not being easily generalizable is not a weakness in a conjecture. In fact, the opposite could be argued.
- l. 11, "have not yielded mathematically valuable results". The paper [8] was published earlier this year. It is too early for mathematically valuable results to have appeared.
- l. 37, "simple polyhedra". The word "simple", in polyhedral theory, has a very well defined meaning (that exactly n edges meet in any vertex of the polyhedron, where n is its dimension), but Euler's relation holds in general. I suggest using the word "convex" in place of "simple".
- l. 40, give at least one reference to the "traditional methods of data-driven conjecture generation".
- l. 63, I find the word "as" here slightly confusing. I suggest substituting it with "namely" or "more precisely".
- ll. 78-79, I would add here a reference to Figure 1, where this process is illustrated.
- ll. 82-83, "it is important ... potentially provable". It would be good to explain here why this is important, and what you mean by "realistic" and "potentially provable".
- footnote 1, "z))" should be "z)))"
- l. 111, "suggesting" seems to me a more appropriate word here than "confirming"
- p. 5, discussion between the Conjecture and the Theorem. Comments as to why \hat{f} "did not care" about $\text{inj}(K)$ would be very welcome here.
- l. 129, the word "very" seems to me more appropriate here than "too"
- l. 133, there is no "Theorem 2" in the manuscript, you could say "The theorem above"
- l. 143, "Irreducible ... analysis". Add a reference here.
- l. 156, the word "representation" here (and in many other places to the end of the paper) is misleading since it is not referring to the mathematical concept but to an encoding of the input data. It would be good to change it.
- p. 7, Fig. 3(a), Figure 3(a) is redundant and misleading since it represents the same data twice (for (i,j) and for (j,i) , which are the same reflection). Delete the lower part of the square, up to and including the diagonal.
- l. 162, "determined the network". Something seems to be wrong here
- l. 162, "analyzing ... compared". This sentence is a bit cryptic. I suggest changing it to "analyzing the edge distribution in these graphs compared".
- l. 164 and ff., "N" and "n" are used interchangeably in this subsection, it

would be good to use just one of them

- l. 167, "not obviously recoverable". I think that a more precise and clear sentence is obtained by deleting the word "obviously" here.

- l. 168, "a Bruhat interval" should be "KL polynomials"

- p. 7, The caption of Figure 4 gives the impression that these substructures were discovered only for the interval shown here. I suggest moving "for the interval ... S_6 " just after the word "Illustration"

- l. 175, "though" should be "through"

- l. 181, the use of "n" here is again misleading. Explain explicitly what the number $3 \cdot 10^6$ refers to)

- l. 195, give at least one reference to the use of machine learning to generate theorems, or change the sentence accordingly.

- l. 197, "and provable". This is a puzzling word to put here. Are the other conjectures "not provable"? How do you know that yours are "provable"? Of course at least one has been proved. In my view you could use the word "deeper" or "deep" in place of either "provable" or "interesting" or in addition to both of them, but I have no strong feelings about this.

- l. 233, "nature" should be "Nature"

- l. 351 "Snappy" should be "SnapPy"

- l. 398, explain what you mean by "non-trivial"

- ll. 406-407, explain what you mean here by "unique" and "size"

- l. 409, "of the" should be "of some of the"

- l. 423, the symbol " $\stackrel{f}{\rightarrow}$ " has not been used up to this point. Is that the same as \hat{f} ? Explain

- p. 14, eq. (4), define "L"

Referee #4 (Remarks to the Author):

This is an extremely interesting paper, very unusual for me to assess as a mathematician.

As requested by the editors, my quick opinion focuses on the advancement within representation theory and how the presented approach compares to "classical" computational methods for assisting mathematical research in this area.

##

The presented advancement concerns Kazhdan-Lusztig polynomials. These are famous and important combinatorial objects in representation theory and geometry. Technically, they are associated to intervals in the Bruhat order on a given Weyl or more generally Coxeter group. One

long-standing conjecture ("combinatorial invariance conjecture") is that, despite their definition does a priori depend on the concrete interval, intervals with isomorphic Bruhat graphs yield equal Kazhdan-Lusztig polynomials.

##

Advancement in representation theory: The concrete advancement for Kazhdan-Lusztig polynomials is a new refinement (which the authors call "hypercube decomposition") together with a concrete combinatorial conjecture concerning this refinement that would imply the combinatorial invariance conjecture. They moreover conjecture the "positivity" of this refinement which is another indication that this decomposition is natural and important.

Such a new approach (even conjectural) to the combinatorial invariance conjecture would certainly be publishable in a top combinatorics journal. In the paper under evaluation, this advancement is merely a use case example of their AI approach to mathematical research.

##

The combinatorics of Coxeter groups and in particular of finite Coxeter groups is perfectly suitable for computer-aided research.

Comparison of the presented AI approach to classical computational methods: The structure of Coxeter groups and their "classifications" makes it often possible to test concrete conjectures on many examples, or even prove a conjecture by implementing it in a computer algebra system and (possibly exhaustively) verify it for (all) "small examples". (The authors of this paper have tested their conjecture for all intervals in symmetric groups on up to 7 letters and also on more than a million intervals in the symmetric groups on 8 or 9 letters.) On the other hand, the quick computer-aided exhibition of interesting (counter-)example makes it possible to promptly discard/modify/refine/generalize conjectures, or to exhibit the "best suited framework" for the situation under consideration.

Classical computer-aided research within the combinatorics of Coxeter groups thus consists of highly intertwined advancement cycles of finding appropriate example (such as a certain type of Coxeter group / of Bruhat interval / of group element / of other combinatorial structures) to exhibit a conjectured property (*), to then test this property on some/many other examples, to discard/modify/refine/generalize the conjecture (***) as much as possible and finally to prove the conjecture using tools suggested by the most appropriate version of the conjecture.

This paper provides a completely new toolbox from AI to be used within this advancement cycle by providing a tool to also introduce the use of computers (and now less so of specialized computer algebra systems but of an---as far as I can judge---generic AI tool) also to the parts (*) and (***) which I consider a crucial part of the "creative process" within research.

##

The paper under consideration appears to be the first approach to mathematical research where an AI approach is used to assist the "creative process" as described above, together with (at least concerning the representation theoretic contribution) a successful "creative contribution" by an AI. I consider and outstanding advancement and strongly suggest publication of this paper in Nature.

Author Rebuttals to Initial Comments:

Referee #1 (Remarks to the Author):

As requested by the editor, my primary role as a referee is to review the significance of the new result in knot theory. For this, I looked at both the main article and the two supplementary files.

The new result, which relates the geometrically defined "natural slope" to the purely topological "signature" is indeed quite striking and unexpected. Independent of its unusual origin, I judge the separate mathematical paper "The signature and cusp geometry of hyperbolic knots" likely to be accepted by the top speciality journal in the field (Geometry and Topology) and be a shoe in at the second best one (J. Topology). For scale, only the top general math journals Annals, JAMS, and Inventiones, which are analogous in math to Nature and Science, have higher standards than Geometry and Topology. So to me the machine learning method of the main paper is an exciting new tool given that it can lead to new results of this quality!

I have two more detailed comments:

1. The initial connection observed between slope and signature is a linear relationship which is extremely marked, as shown in Figure 2(b) of the main submission and especially Figure 5 of "Supplementary Material: The signature and cusp shape of hyperbolic knots". It would strengthen the case for using machine learning if you explained why this wouldn't have been noticed by a quick correlation check of the variables or a naive linear regression.

We've added some sentences clarifying that while the final relationship with slope is linear, that is because we've defined slope in such a way and that the relationship with the original quantities is multivariate and non-linear.

2. It might be worth mentioning some prior instances where people have experimentally explored connections between geometric and topological invariants of knots, specifically:

Khovanov, Patterns in Knot Cohomology, I.

<http://projecteuclid.org/euclid.em/1087329238>

Jejja et al, Deep learning the hyperbolic volume of a knot

<https://doi.org/10.1016/j.physletb.2019.135033>

Unlike the current paper, I believe these only led to some interesting (and still unexplained) patterns and not a new theorem as in the paper under review, which highlights the strength of the new method.

These have been added and thanks especially for pointing us to the Khovanov paper.

Referee #2 (Remarks to the Author):

A Summary of the key results) Summary of paper: It tries to enhance and formalize the idea of discovering new mathematical relationships via computer experimentation. For a mathematical object X (which is a vector of all known attributes or quantitative facts about the object) we wish to understand a quantity $Y(X)$ and understand how it relates to various parts of X . The setup requires some way to generate objects X and compute $Y(X)$. The key ingredients are: (a) using examples of $(X, Y(X))$ learn via supervised machine learning a way to predict $Y(X)$ from X (b) use saliency/attribution methods to learn portions of X that determine $Y(X)$ and (c) derive —again via machine learning— a relationship between these. (Aside: maybe the methodology should include (c) after line 47?)

At first sight one might fear that if one is able to compute $Y(X)$ already via a separate algorithm then perhaps one already understands $Y(\cdot)$ pretty well already? If so, how interesting would be the new insights? However, the authors present two examples of new relationships found in two areas of mathematics (knot theory and representation theory) via this method which --I presume---were surprising to the mathematicians.

B Originality and significance) Given the increasing interest in use of machine learning (and more general computer science ideas) to advancing mathematics, it is timely and significant.

In terms of originality, this project clearly required close collaboration between experts from machine learning and mathematics and I am not aware of any other competing results or efforts.

(C Data & methodology) The paper is well-written and the techniques are sound and of general interest. I verified the details of machine learning and they seem solid.

The paper opens the door for more work of this type. The machine learning component is fairly simple and probably can be carried out by mathematicians who know any sort of computer programming.

(D Appropriate use of statistics and treatment of uncertainties) N/A. (There are no statistical theory applicable to this merger of machine learning and human intuition.)

(E Conclusions: robustness, validity, reliability) See above

(F Suggested improvements: experiments, data for possible revision) Suggestions:

1) The main paper could give a bit of insight to the average reader about attribution methods, since the math is fairly simple. (Namely, Gradient captures how $Y()$ changes as we change coordinates of X). Maybe modify the opening example of Euler's formula, but assume X contains "irrelevant" information about the polyhedron. Then the method has to first infer that only number of vertices and edges is relevant information.

That's an excellent point, we've modified the example slightly to include the "irrelevant" information of surface area and volume, which makes it much more effective. We've also added a little more about gradients in both the main section and methods

2) Most readers will not be experts in knot theory and representation theory, and it would be helpful to indicate how interesting the discovered facts are for experts in these areas. Are these something that an average grad student could have discovered? How about an average math professor?

While we would like to clarify this in the paper, It's hard for us to objectively make this claim. We believe both are results at the forefront of mathematics research.

(3) Include other interesting settings where $Y(X)$ is computable from X , but we don't fully understand function $Y()$. Since the author list includes some top mathematicians I assume there have been internal discussions of this issue.

This is a really interesting setting! While we have discussed it, we'd prefer to leave this as an open question for the moment, so have mentioned it as such in the methods.

(G References: appropriate credit to previous work?) Reference list is good. It might be helpful to briefly mention various ways in which computer science methods are being applied to mathematics and to mention how this work does not fit in those traditions (eg formal verification).

(H Clarity and context: lucidity of abstract/summary, appropriateness of abstract, introduction and conclusions) See above.

Overall I am positive about the paper.

Referee #3 (Remarks to the Author):

Review of

Advancing mathematics by guiding human intuition with AI

by A. Davies and al.

As I am not an expert in knot theory, nor in AI, I will comment mainly on the representation theory part of the work. I find no flaws in the manuscript which should prohibit its publication.

The mathematical conclusions of this work are original. The representation theory results presented are of immediate interest to many people in representation theory, but also in algebraic geometry and topology, given that Kazhdan-Lusztig polynomials play such a central role in representation theory, but also have applications to the algebraic geometry and topology of Schubert varieties (see, e.g., [A. Bjorner, F. Brenti, Combinatorics of Coxeter Groups, Graduate Texts in Mathematics, 231, Springer-Verlag, New York, 2005], [S. Billey, V. Lakshmibai, Singular loci of Schubert varieties, Progress in Mathematics, 182, Birkhäuser Boston, Inc., Boston, MA, 2000], [J. E. Humphreys, Representations of semisimple Lie algebras in the BGG category O , Graduate Studies in Mathematics, vol. 94, Amer. Math. Soc., Providence, RI, xvi+289 pp., 2008], and the references cited

there) and that the conjecture studied (the combinatorial invariance conjecture) is now probably the most outstanding

open problem about these polynomials, and has been open for about 40 years ([31]). The methodology presented of using AI in mathematical research is of immediate interest to any mathematician who works with mathematical objects that can be conveniently stored and analyzed by a computer. This includes mathematicians working, for example, in combinatorics, algebra, algebraic geometry, and many parts of topology and mathematical physics.

I cannot imagine such a mathematician not using these methods, where available, given that not using them means progressing much more slowly and being unable to analyze very complicated objects and very large datasets. I cannot comment on the validity of the AI method. However, the mathematical results obtained with it are quite impressive, which makes me think that the method is definitely very valid. The authors have done an exceedingly good job of presenting the material to a general scientific audience. The whole paper, including the abstract, is clear and understandable, with a few exceptions noted below. The reporting of data and methodology is sufficiently detailed for a computer scientist to reproduce the results, but I strongly suggest adding quite a few more details (at least a few lines) for a mathematician to be able to do so. Error bars in graphs and tables are accurately described and easy to understand. As I am not an expert in statistics, I cannot comment on the

appropriateness of the statistical tests performed. I think that the conclusion of the paper, namely that AI can be an extremely useful tool in mathematical research, is valid and has been very convincingly demonstrated in this work.

This paper marks the beginning of a new phase in the use of computers in mathematical research. While up to now computers have been used to test existing conjectures or, lately, to find new conjectures, this is the first time that computers have been able to suggest avenues of proof for existing conjectures. This has been achieved by allowing humans and deep learning algorithms to interact in a new way, which I consider to be one of the two outstanding features of this work. The other outstanding feature is the quality, novelty, depth, and interest of the conjectures and theorems produced with this method. In particular, the feeling that extremal reflections seem to play a crucial role in the conjectured combinatorial invariance of KL polynomials is one that I would have expected only from a handful of great world experts in representation theory, and is such a "human" suggestion that it gives me the goose bumps. For all the above reasons, I warmly recommend the publication of

this paper in Nature. The authors conclude their manuscript with the sentence: "More broadly, it is our hope that this framework is an effective mechanism to allow for the introduction of machine learning into mathematicians' work, and encourage further collaboration between the two fields." I can't but join them in their hope.

Detailed comments:

- l. 10 "in ways that do not easily generalize". I feel that not being easily generalizable is not a weakness in a conjecture. In fact, the opposite could be argued.

This sentence was unclear - it was intended to mean that the techniques don't easily generalize. This has been rephrased to improve clarity

- l. 11, "have not yielded mathematically valuable results". The paper [8] was published earlier this year. It is too early for mathematically valuable results to have appeared.

Amended this to state that they have not yet yielded such results

- l. 37, "simple polyhedra". The word "simple", in polyhedral theory, has a very well defined meaning (that exactly n edges meet in any vertex of the polyhedron, where n is its dimension), but Euler's relation holds in general. I suggest using the word "convex" in place of "simple".

Thanks, done.

- l. 40, give at least one reference to the "traditional methods of data-driven conjecture generation".

Done

- l. 63, I find the word "as" here slightly confusing. I suggest substituting it with "namely" or "more precisely".

Done

- ll. 78-79, I would add here a reference to Figure 1, where this process is illustrated.

Done

- ll. 82-83, "it is important ... potentially provable". It would be good to explain here why this is important, and what you mean by "realistic" and "potentially provable".

Thank you for pressing for clarity here, we have replaced with “In this process, the mathematician can guide the choice of conjectures to those that not just fit the data but also seem interesting, plausibly true and ideally, suggestive of a proof strategy”.

- footnote 1, "z))" should be "z)))"

Done

- l. 111, "suggesting" seems to me a more appropriate word here than "confirming"

Done

- p. 5, discussion between the Conjecture and the Theorem. Comments as to why \hat{f} "did not care" about $\text{inj}(K)$ would be very welcome here.

This is an excellent point - the attribution in fact does highlight both the injectivity radius and the shortest geodesic and the paragraph has been rephrased to make this clear.

- l. 129, the word "very" seems to me more appropriate here than "too"

Done

- l. 133, there is no "Theorem 2" in the manuscript, you could say "The theorem Above"

Done

- l. 143, "Irreducible ... analysis". Add a reference here.

Done

- l. 156, the word "representation" here (and in many other places to the end of the paper) is misleading since it is not referring to the mathematical concept but to an encoding of the input data. It would be good to change it.

This is a very good point, done.

- p. 7, Fig. 3(a), Figure 3(a) is redundant and misleading since it represents the same data twice (for (i,j) and for (j,i) , which are the same reflection). Delete the lower part of the square, up to and including the diagonal.

Done.

- l. 162, "determined the network". Something seems to be wrong here

Fixed

- l. 162, "analyzing ... compared". This sentence is a bit cryptic. I suggest changing it to "analyzing the edge distribution in these graphs compared".

Done

- l. 164 and ff., "N" and "n" are used interchangeably in this subsection, it would be good to use just one of them

Done

- l. 167, "not obviously recoverable". I think that a more precise and clear sentence is obtained by deleting the word "obviously" here.

That's correct, done.

- l. 168, "a Bruhat interval" should be "KL polynomials"

Thanks, done.

- p. 7, The caption of Figure 4 gives the impression that these substructures were discovered only for the interval shown here. I suggest moving "for the interval ... S₆" just after the word "Illustration"

Done

- l. 175, "though" should be "through"

Done

- l. 181, the use of "n" here is again misleading. Explain explicitly what the number $3 \cdot 10^6$ refers to)

Done

- l. 195, give at least one reference to the use of machine learning to generate theorems, or change the sentence accordingly.

Removed

- l. 197, "and provable". This is a puzzling word to put here. Are the other conjectures "not provable"? How do you know that yours are "provable"? Of course at least one has been proved. In my view you could use the word "deeper" or "deep" in place of either "provable" or "interesting" or in addition to both of them, but I have no strong feelings about this.

Amended as suggested.

- l. 233, "nature" should be "Nature"

Done

- l. 351 "Snappy" should be "SnapPy"

Done

- l. 398, explain what you mean by "non-trivial"

Done

- ll. 406-407, explain what you mean here by "unique" and "size"

Done

- l. 409, "of the" should be "of some of the"

Done

- l. 423, the symbol " $\stackrel{f}{\rightarrow}$ " has not been used up to this point.

Is that the same as \hat{f} ? Explain

It should indeed be \hat{f} , thanks.

- p. 14, eq. (4), define "L"

Done

Referee #4 (Remarks to the Author):

This is an extremely interesting paper, very unusual for me to assess as a mathematician.

As requested by the editors, my quick opinion focuses on the advancement within representation theory and how the presented approach compares to "classical" computational methods for assisting mathematical research in this area.

##

The presented advancement concerns Kazhdan-Lusztig polynomials. These are famous and important combinatorial objects in representation theory and geometry. Technically, they are associated to intervals in the Bruhat order on a given Weyl or more generally Coxeter group. One long-standing conjecture ("combinatorial invariance conjecture") is that, despite their definition does a priori depend on the concrete interval, intervals with isomorphic Bruhat graphs yield equal Kazhdan-Lusztig polynomials.

##

Advancement in representation theory: The concrete advancement for Kazhdan-Lusztig polynomials is a new refinement (which the authors call "hypercube decomposition") together with a concrete combinatorial conjecture concerning this refinement that would imply the combinatorial invariance conjecture. They moreover conjecture the "positivity" of this refinement which is another indication that this decomposition is natural and important.

Such a new approach (even conjectural) to the combinatorial invariance conjecture would certainly be publishable in a top combinatorics journal. In the paper under evaluation, this advancement is merely a use case example of their AI approach to mathematical research.

##

The combinatorics of Coxeter groups and in particular of finite Coxeter groups is perfectly suitable for computer-aided research.

Comparison of the presented AI approach to classical computational methods: The structure of Coxeter groups and their "classifications" makes it often possible to test concrete conjectures on many examples, or even prove a conjecture by implementing it in a computer algebra system and (possibly exhaustively) verify it for (all) "small examples". (The authors of this paper have tested their conjecture for all intervals in symmetric groups on up to 7 letters and also on more than a million intervals in the symmetric groups on 8 or 9 letters.) On the other hand, the quick computer-aided exhibition of interesting (counter-)example makes it possible to promptly discard/modify/refine/generalize conjectures, or to exhibit the "best suited framework" for the situation under consideration.

Classical computer-aided research within the combinatorics of Coxeter groups thus consists of highly intertwined advancement cycles of finding appropriate example (such as a certain type of Coxeter group / of Bruhat interval / of group element / of other combinatorial structures) to exhibit a conjectured property (*), to then test this property on some/many other examples, to discard/modify/refine/generalize the conjecture (**), as much as possible and finally to prove the conjecture using tools suggested by the most appropriate version of the conjecture.

This paper provides a completely new toolbox from AI to be used within this advancement cycle by providing a tool to also introduce the use of computers (and now less so of specialized computer algebra systems but of an---as far as I can judge---generic AI tool) also to the parts (*) and (**) which I consider a crucial part of the "creative process" within research.

##

The paper under consideration appears to be the first approach to mathematical research where an AI approach is used to assist the "creative process" as described above, together with (at least concerning the representation theoretic contribution) a successful "creative contribution" by an AI. I consider and outstanding advancement and strongly suggest publication of this paper in Nature.